# Study on Cutting Performance of Micro Groove Tool in Turning AISI 304 and Surface Quality of the Workpiece

**Zhenghong Liu [1] and Jinxing Wu [2],***

1 School of Mechanical Engineering, Guiyang University, Guiyang 550005, China
2 School of Mechanical and Electrical Engineering, Gui Zhou Min Zu University, Guiyang 550025, China
* Correspondence: gs.jxwu16@gzu.edu.cn

**Abstract:** AISI 304 has high-tensile strength and excellent corrosion resistance, which is widely needed in the energy industry and equipment manufacturing industry. However, the tools for cutting AISI 304 are easy to wear and have short service life. In order to improve tool life, micro grooves are designed on the rake face of the tool for the machining of AISI 304. Through the single factor cutting experiment, it is found that under the same cutting parameters, the micro groove tool has less cutting depth resistance than the initial tool; the main cutting force and feed resistance are reduced by more than 15%. The shear energy is reduced by more than 13%; the surface roughness and the hardening degree of the workpieces are reduced. Through the durability test, it is found that the service life of the micro groove tool is 57% longer than that of the initial tool, and the abrasive wear, bonding wear, and oxidation wear of the tool are significantly less. Through cutting experiments and theoretical analysis, the cutting performance of the micro groove tool has been improved.

**Keywords:** micro groove tool; cutting force; energy; wear; surface quality





## 1. Introduction

AISI 304 has good comprehensive mechanical properties and is widely used in the field of equipment manufacturing. However, due to its good plasticity and high toughness, the tool wear is faster in the process of cutting; hence, the service life of the tool is short, and the use cost of the tool is higher. To solve this problem, the paper has carried out an innovative optimization design of the tool to reduce the cutting force and tool wear and improve the service life of the tool.

Tool micro texture design is an important subject of tool optimization design. Duong et al. [1–3] verified the feasibility of the micro texture tool through numerical analysis and verified by relevant experiments. Deng et al. [4] used femtosecond laser technology to prepare a groove-shaped micro texture on the surface of cemented carbide tools and deposited tungsten disulfide solid lubricating coating on the rake face of texture tool. The influence of texture tool on cutting performance during dry cutting was studied by using three tools, including the original tool, the texture tool with solid lubricant deposited, and the texture tool without solid lubricant. The results show that the cutting temperature, cutting force [5], and the friction coefficient of the tool chip interface of the texture tool are significantly lower than that of the original tool, and the texture tool with tungsten disulfide solid lubricating coating is the best in improving the cutting performance. Micro texture can improve the friction characteristics between two friction pairs [6–8]. Micro-texture can change the friction conditions between tool and workpiece [9]. Sugihara et al. [10–12] prepared different textures on the rake face of the tool and studied the effect of texture on the rake face during cutting. The results show that the micro texture can reduce the friction between the tool and the chip and reduce the wear of the rake face of the tool. Johannes et al. [13,14] designed different micro textures to improve the adhesion between the tool and the chip and found that the textured tool has a good effect on reducing the

adhesion behavior and the tool chip friction. Micro-textured tools perform well in reducing tool wear [15].

The surface quality of the workpieces is directly related to the usability and reliability of workpieces. Yang et al. [16] used a micro texture milling tool to mill titanium alloy and found that the micro texture tool can improve the hardening degree of the titanium alloy surface, especially if the surface roughness of the workpiece has a high correlation with the diameter and depth of the micro texture. Wang et al. [17] applied the rounded micro groove tool to cut composite materials and found that the micro groove tool can improve the surface quality of the workpiece. Pan et al. [18] prepared a linear micro groove texture and a V-shaped micro groove texture on the rake face of the milling tool. Through milling experiments, it is shown that both textures can reduce the surface roughness of the workpiece, and the surface roughness of the V-shaped micro groove texture decreases more. Zhang et al. [19] compared the cutting performance of textured tools and traditional tools in cutting AISI 1045. The experimental results show that the micro texture of the tool can effectively reduce the surface roughness of the workpiece, and the effect is better under lubrication conditions. When cutting 304 stainless steel, Ahmed et al. [20] found that the surface roughness of the workpiece processed by the square texture tool is smaller, and the surface hardening degree of the workpiece is lower.

Many studies show that the micro textured tool can reduce the contact length between the tool and the chip and reduce the abrasive wear, adhesive wear, and oxidation wear caused by the contact stress and high temperature of the tool [21–23]. Singh et al. [24] studied the wear behavior of textured tools when cutting Ti6Al4V in different environments and found that the mixed graphene lubrication conditions, the crater wear, and the bond wear of the textured tools decreased. Liu et al. [25] studied the protective effect of nano texture on the non-worn surface and pointed out that nano texture reduces the flank wear of the tool. Sarvesh et al. [26] found that the micro texture tool reduces the friction coefficient and reduces the abrasive wear of the tool. Yang et al. [27] used laser processing to prepare micro texture on the rake face of cemented carbide tools and used micro texture tools to carry out milling titanium alloy to study the effect of micro texture parameters on the wear of tool rake face. The results show that the micro texture parameters have a significant impact on the wear of the tool rake face. Fang et al. [28] evaluated the effect of micro texture on tool wear and adhesion during longitudinal turning of Inconel 718. The experimental results show that micro textured cutting tools can generally reduce the flank wear and crater wear compared with non-textured cutting tools. Musavi et al. [29] carried out cutting experiments of the original tool and the micro groove tool under MQL conditions. The result showed that the cutting performance of the micro groove tool was better than that of the original tool. It was found that the groove spacing had the greatest impact on the tool wear and surface roughness.

The above research shows that the micro texture design can improve the comprehensive cutting performance of the tool, but few researchers study the cutting performance of the micro texture tool in different cutting parameters, which can provide useful data for many factories. In this paper, the cutting performance of the micro groove tool is analyzed under different cutting parameters. At the same time, the tool durability experiment was carried out to study the tool wear.

## 2. Materials and Experimental Process

The tools used in this paper are provided by a manufacturer, which is called tool A for short. The tool with micro grooves on the rake face is called tool B, and both tools A and B are cemented carbide. The workpieces to be processed are AISI 304. Tools A and B have the same geometric angle, as shown in Table 1.

**Table 1.** Geometric angles of the tools.

| Geometric Angle | Tool Angle $\varepsilon_r$ | Rake Angle $\gamma_0$ | Flank Angle $\alpha_0$ | Main Edge Angle $K_r$ | End Edge Angle $K_r'$ | Inclination Angle $\lambda_s$ |
|---|---|---|---|---|---|---|
| Value (°) | 80 | 8 | 7 | 95 | −5 | −5 |

First, through the special cutting simulation software (version Deform 11.0), Deform, the cutting AISI 304 simulation is carried out by tool A; the cutting parameters are: cutting speed, v = 120 m/min, feed rate, f = 0.15 mm, and cutting depth, ap = 1.5 mm. Through the cutting simulation test, the design of tool B is carried out based on the theory of temperature field [30,31]. Both tools A and B are cemented carbide with WC as the main body, and their surfaces are coated with TiAlN coating. The matrix element content of AISI 304 is shown in Table 2.

**Table 2.** Chemical composition of AISI 304 (wt%).

| Si | Mn | P | S | Ni | Cr | C |
|---|---|---|---|---|---|---|
| | | | | | | Fe |
| 0.75 | 1.64 | 0.045 | 0.03 | 8.56 | 18.87 | 0.08 |
| | | | | | | 70.025 |

*2.1. Three-Dimensional Cutting Force Analysis*

Within the range of cutting parameters and based on the needs of actual production, each enterprise will choose different cutting parameters. Based on the combination of cutting parameters of most enterprises cutting AISI 304 and taking into account the operability of the experiment, the paper designs the experiment of single factor cutting parameter changes of cutting AISI 304 with tools A and B.

The cutting experiment was carried out on the C2-6136hk CNC lathe (the favgol, Chongqing, China). In the cutting process, the three-dimensional cutting force of the tool was tested by the force measuring instrument, Kisler-9257b (kisler-9257-b, kisler Instrumente AG, Winterthur, Switzerland). Tools A and B and the test instruments are shown in Figure 1.

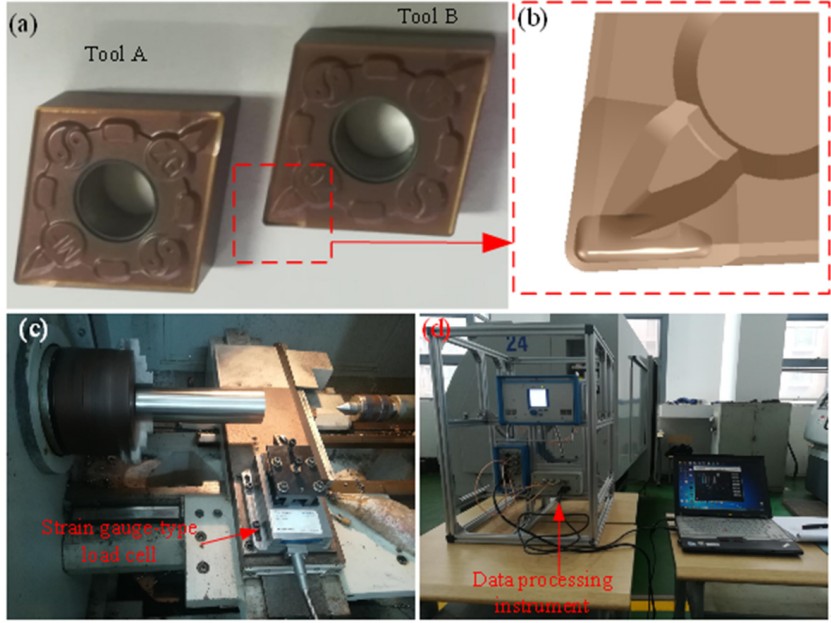

**Figure 1.** (**a**) Tools A and B, (**b**) micro groove morphology, (**c**) NC experimental platform, (**d**) data processing device.

The total length of the workpiece bar is 220 mm. During the single factor experiment, the cutting length of each parameter combination is 15 mm, and the cutting is carried out continuously until the experiment is completed. The experimental design scheme is shown in Tables 3 and 4, which contain the measured cutting force values.

**Table 3.** Three-dimensional cutting force value of the tool A.

| Variation of Cutting Force of Tool A with Cutting Parameters | | | | | |
|---|---|---|---|---|---|
| $V_c$ (m/min) | f (mm/r) | $a_p$ (mm) | $F_x$ (N) | $F_y$ (N) | $F_z$ (N) |
| 100 | 0.15 | 1.5 | 103 | 512 | 283 |
| 120 | 0.15 | 1.5 | 110 | 530 | 290 |
| 140 | 0.15 | 1.5 | 114 | 562 | 302 |
| 160 | 0.15 | 1.5 | 121 | 578 | 315 |
| 180 | 0.15 | 1.5 | 129 | 628 | 327 |
| 120 | 0.11 | 1.5 | 103 | 501 | 252 |
| 120 | 0.13 | 1.5 | 108 | 516 | 271 |
| 120 | 0.15 | 1.5 | 110 | 530 | 290 |
| 120 | 0.17 | 1.5 | 115 | 549 | 334 |
| 120 | 0.19 | 1.5 | 117 | 564 | 365 |
| 120 | 0.15 | 1.1 | 83 | 497 | 274 |
| 120 | 0.15 | 1.3 | 95 | 510 | 280 |
| 120 | 0.15 | 1.5 | 110 | 530 | 290 |
| 120 | 0.15 | 1.7 | 125 | 551 | 302 |
| 120 | 0.15 | 1.9 | 137 | 589 | 324 |

**Table 4.** Three-dimensional cutting force value of the tool B.

| Variation of Cutting Force of Tool B with Cutting Parameters | | | | | |
|---|---|---|---|---|---|
| $V_c$ (m/min) | f (mm/r) | $a_p$ (mm) | $F_x$ (N) | $F_y$ (N) | $F_z$ (N) |
| 100 | 0.15 | 1.5 | 89 | 461 | 204 |
| 120 | 0.15 | 1.5 | 90 | 488 | 216 |
| 140 | 0.15 | 1.5 | 93 | 517 | 243 |
| 160 | 0.15 | 1.5 | 98 | 532 | 262 |
| 180 | 0.15 | 1.5 | 98 | 567 | 268 |
| 120 | 0.11 | 1.5 | 86 | 463 | 190 |
| 120 | 0.13 | 1.5 | 88 | 476 | 204 |
| 120 | 0.15 | 1.5 | 90 | 488 | 216 |
| 120 | 0.17 | 1.5 | 90 | 514 | 232 |
| 120 | 0.19 | 1.5 | 94 | 537 | 272 |
| 120 | 0.15 | 1.1 | 80 | 451 | 204 |
| 120 | 0.15 | 1.3 | 85 | 462 | 210 |
| 120 | 0.15 | 1.5 | 90 | 488 | 216 |
| 120 | 0.15 | 1.7 | 97 | 498 | 221 |
| 120 | 0.15 | 1.9 | 115 | 524 | 241 |

To comprehensively compare the cutting performances of tools A and B under the same conditions and more clearly describe the change of the three-dimensional cutting force of tools with cutting parameters, this paper draws the change diagram of cutting force with cutting speed, feed speed, and cutting depth, as shown in Figures 2–4.

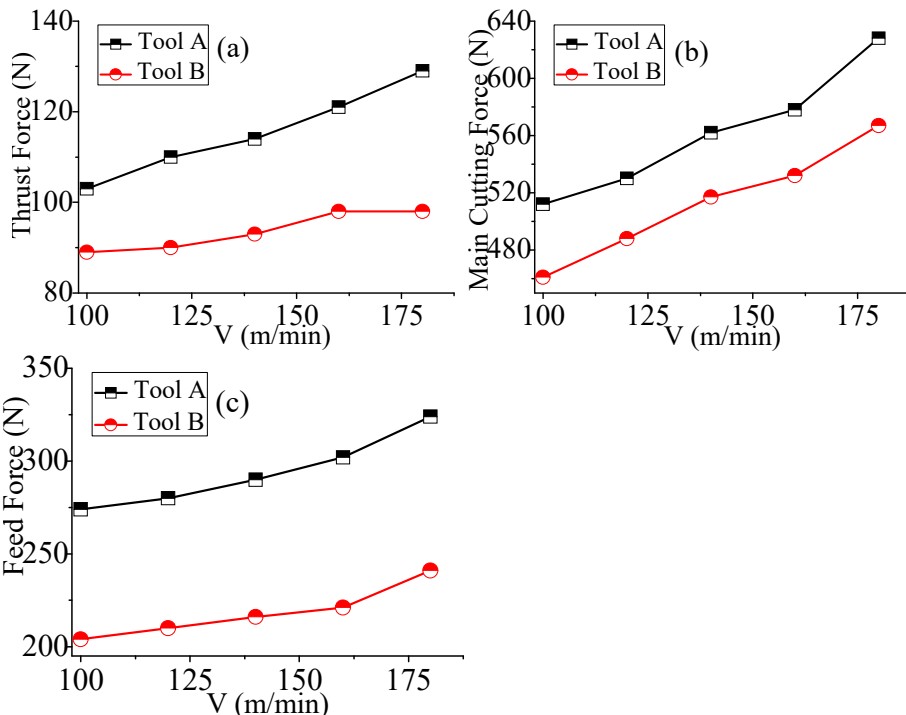

**Figure 2.** (**a**) Cutting depth resistance changes with cutting speed, (**b**) main cutting force changes with cutting speed, (**c**) feed resistance changes with cutting speed.

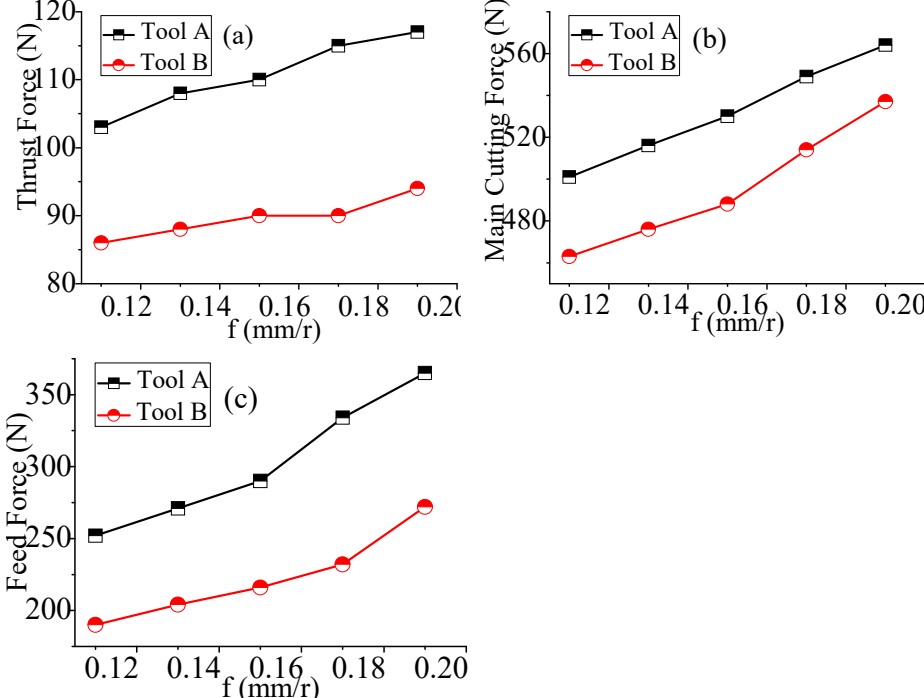

**Figure 3.** (**a**) Cutting depth resistance changes with feed speed, (**b**) main cutting force changes with feed speed, (**c**) feed resistance changes with feed speed.

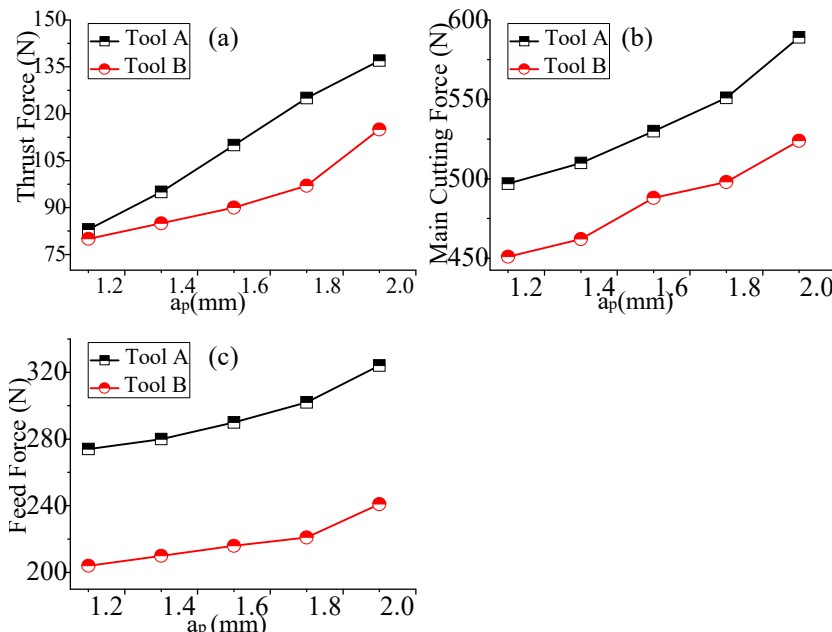

**Figure 4.** (**a**) Cutting depth resistance changes with cutting depth rate, (**b**) main cutting force changes with cutting depth rate, (**c**) feed resistance changes with cutting depth rate.

As shown in Figure 2a, when the feed rate and cutting depth rate remain unchanged, the cutting depth resistances of the tools A and B increase with the increase of cutting speed, and the cutting depth resistance of the tool B changes more smoothly with the increase of cutting speed. The cutting depth resistance of the tool A is slightly higher than that of the tool B. It can be seen from Figure 2b that the main cutting forces of tools A and B increase with the increase of cutting speed. At the same cutting speed, the main cutting force of the tool A is more than 10% higher than that of the tool B. It can be seen from Figure 2c that the feed resistance of the tools A and B increase with the increase of the cutting speed. Under the same conditions, the tool B has smaller feed resistance.

As shown in Figure 3a, when the cutting speed and cutting depth rate remain unchanged, the cutting depth resistance of the tool A increases with the increase of feed speed, and the cutting depth resistance of the tool B increases slowly with the increase of feed speed. In general, the cutting depth resistance of the tool A is higher than that of the tool B. It can be seen from Figure 3b that the main cutting forces of the tools A and B increase with the increase of the feed speed. At the same feed speed, the main cutting force of the tool B is reduced by more than 10% compared with the tool A. It can be seen from Figure 3c that the feed resistances of the tools A and B increase with the increase of the feed speed. At the same feed speed, the feed resistance of the tool B is more than 10% lower than that of the tool A.

As shown in Figure 4a, when the cutting speed and feed rate remain unchanged, the cutting depth resistances of the tools A and B increase with the increase of the cutting depth rate. On the whole, the cutting depth resistance of the tool A is slightly higher than that of the tool B. It can be seen from Figure 4b that the main cutting forces of the tools A and B increase with the increase of the feed speed. Under the same conditions, the main cutting force of the tool A is at least 10% higher than that of the tool B. It can be seen from Figure 4c that the feed resistance of the tools A and B increase with the increase of the cutting depth rate. Under the same conditions, the feed resistance of the tool A is more than 25% than that of the tool B.

## 2.2. Research on Cutting Energy

In the process of metal cutting, the chip is deformed, broken, and separated by continuous energy input. The total cutting energy includes: the shear energy in the

first deformation zone, the friction energy in the second deformation zone, the surface energy of the newly formed surface of the cutting workpiece, and the change energy generated when the material passes through the shear surface. Generally, the proportion of motion change energy and surface energy is very small, which is generally ignored in the calculation process. Therefore, the paper mainly calculates the shear energy and friction energy. According to the metal cutting theory and relevant geometric knowledge [31], the following formula can be obtained:

$$\tan \gamma_n = \tan \gamma_0 \cos \gamma_s \tag{1}$$

$$\tan \varphi_n = \tan \frac{a/a_c \cos \gamma_n}{1 - a/a_c \sin \gamma_n} \tag{2}$$

$$\cos \eta_c = t_c \cos \lambda_s / a_w \tag{3}$$

$$\tan \eta_s = \frac{\tan \lambda_s \cos(\varphi_n - \gamma_n) - \tan \eta_c \sin \varphi_n}{\cos \gamma_n} \tag{4}$$

$$v_c = \frac{\sin \phi_n \cos \lambda_s}{\cos v_c \cos(\phi_n - \gamma_n)} \tag{5}$$

$$v_s = \frac{\cos \phi_n \cos \lambda_s}{\cos v_s \cos(\phi_n - \gamma_n)} \tag{6}$$

where $\lambda_s$, $\varphi_n$, $a$, $a_c$, $t_c$, and $a_w$ are the inclination angle, normal shear angle, cutting thickness, chip thickness, chip width, and uncut width, respectively, $\gamma_0$ and $\gamma_n$ are the rake angle and normal angle, respectively, and $\eta_c$ and $\eta_s$ indicate chip flow angle and shear chip flow angle, respectively. The three-dimensional model and two-dimensional cutting model can be equivalently transformed by the angle transformation. To obtain the force of the three-dimensional coordinate system, you can first rotate the angle, $\gamma_n$, around the axis, $\chi'$, and then rotate the angle, $\lambda_s$, around the $z$ axis:

$$\begin{pmatrix} F'_x \\ F'_y \\ F'_z \end{pmatrix} = \begin{pmatrix} 1 & 0 & 0 \\ 0 & \cos \gamma_n & \sin \gamma_n \\ 0 & -\sin \gamma_n & \cos \gamma_n \end{pmatrix} \begin{pmatrix} \cos \lambda_s & -\sin \lambda_s & 0 \\ \sin \lambda_s & \cos \lambda_s & 0 \\ 0 & 0 & 1 \end{pmatrix} \begin{pmatrix} F_x \\ F_y \\ F_z \end{pmatrix} \tag{7}$$

Then the friction force and normal force on the rake face of the tool are respectively:

$$\begin{aligned} F_f &= \sqrt{F_x'^2 + F_z'^2} \\ &= \sqrt{\left((F_x \cos \lambda_s - F_y \sin \lambda_s)^2 + (-F_x \sin \lambda_s \sin \gamma_n - F_y \cos \lambda_s \sin \gamma_n + F_z \cos \gamma_n)^2\right)} \end{aligned} \tag{8}$$

$$F_n = F_x \sin \lambda_s \cos \gamma_n + F_y \sin \gamma_n + F_z \cos \lambda_s \cos \gamma_n \tag{9}$$

The shear force on the shear surface is:

$$F_s = \left[(F_x \cos \lambda_s - F_z \sin \lambda_s)^2 + (F_x \sin \lambda_s \cos \phi_n - F_y \sin \phi_n + F_z \cos \lambda_s \cos \phi_n)^2\right]^{1/2} \tag{10}$$

In the cutting process, the cutting energy is mainly consumed by the shear energy, $N_{ss}$, and friction energy, $N_{sf}$, on the shear surface. Their calculation formula is as follows:

$$N_{ss} = \frac{F_s v_s}{v a_w a} = \tau_s \frac{v_s}{v \sin \phi} \tag{11}$$

$$N_{sf} = \frac{F_f v_c}{v a_w a} = \frac{F_s}{\xi a_w a} \tag{12}$$

In the process of the cutting experiment, the shear energy and friction energy can be calculated by measuring the three-dimensional cutting force and chip thickness, combined with the known parameters and Formulas (1) to (12) [31].

As shown in Figure 5a, when the feed rate and cutting depth rate remain unchanged, the shear energy of the tools A and B increase with the increase of the cutting speed. Under the same cutting speed, the shear energy of the tool B decreases by more than 10% compared to tool A. It can be seen from Figure 5b that the shear energy of the tools A and B increase with the increase of the feed speed. Under the same conditions, the shear energy of the tool B decreases by about 8% compared to tool A. It can be seen from Figure 5c that the shear energy of the tools A and B increase with the increase of the cutting depth rate, and the shear energy of the tool B is smaller under the same conditions. It can be seen from the above that the shear energy of the tool B with different cutting parameters is smaller than that of the tool A.

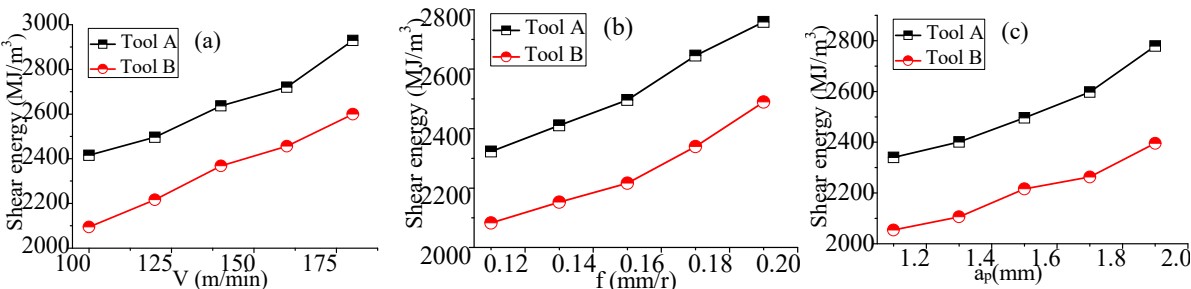

**Figure 5.** (**a**) Shear energy changes with cutting speed, (**b**) Shear energy changes with cutting feed speed, (**c**) Shear energy changes with depth rate.

It can be seen from Figure 6a that the friction energy of the tools A and B increase with the increase of the cutting speed. Under the same conditions, the friction energy of the tool A decreases by more than 8% compared to tool A. It can be seen from Figure 6b that the friction energy of the tools A and B increase with the increase of the feed speed. Under the same conditions, the friction energy of the tool B decreases by about 7%. It can be seen from Figure 6c that the friction energy of the tools A and B increase with the increase of the cutting depth rate, and the friction energy of the tool B is smaller under the same conditions. In general, the friction energy of tool A with different cutting parameters is smaller than that of the tool A.

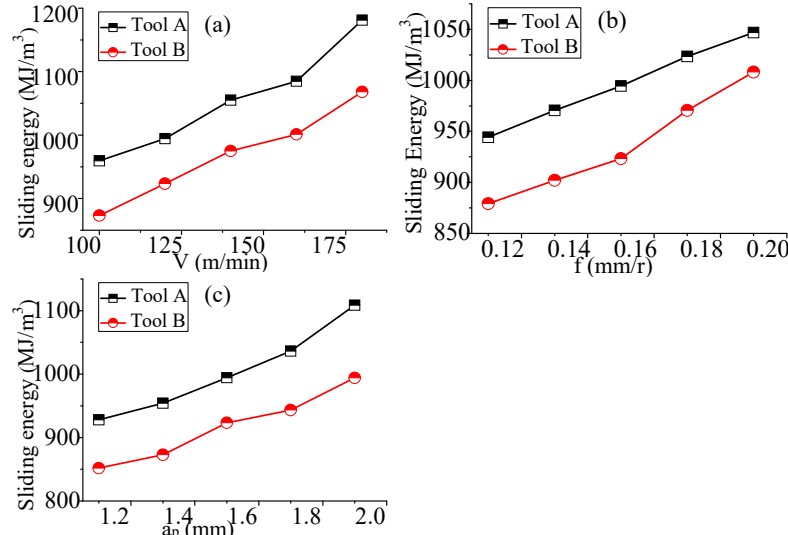

**Figure 6.** (**a**) Friction energy changes with cutting speed, (**b**) friction energy changes with feed speed, (**c**) friction energy changes with cutting depth rate.

In the cutting process, the cutting energy input of tool B is reduced due to the placement of micro grooves, which shows better energy consumption performance.

### 2.3. Workpiece Surface Roughness Analysis

Surface roughness is a micro geometric feature with small spacing of less than 1 mm formed on the surface of the workpiece due to the interaction between the tool and the workpiece in the cutting process. It is an important technical index to evaluate the machining quality. Surface roughness not only affects the wear resistance, fatigue resistance, and sealing performance of the parts, but also affects the service life of the workpiece. The surface roughness measurement and calculation need to pay attention to many factors, such as measurement noise and uncertainty evaluation [32–35]. The surface roughness is measured by Mahr made in Germany, as shown in Figure 7.

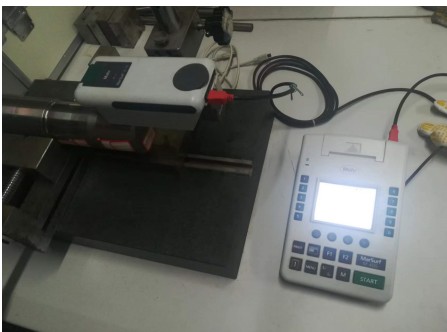

**Figure 7.** Surface roughness measurement.

It can be seen from Figure 8a that when the feed rate and cutting depth remain unchanged, the surface roughness of the workpiece by the tool A increases first, then decreases, and then increases rapidly with the increase of cutting speed. The surface roughness of the workpiece by the tool B increases with the increase of cutting speed. When the cutting speed is 140 m/min, the surface roughness of the workpiece by the tool B is slightly larger than that of the tool A. Under other speed conditions, the surface roughness of the workpiece by the tool B is less than that of the tool A.

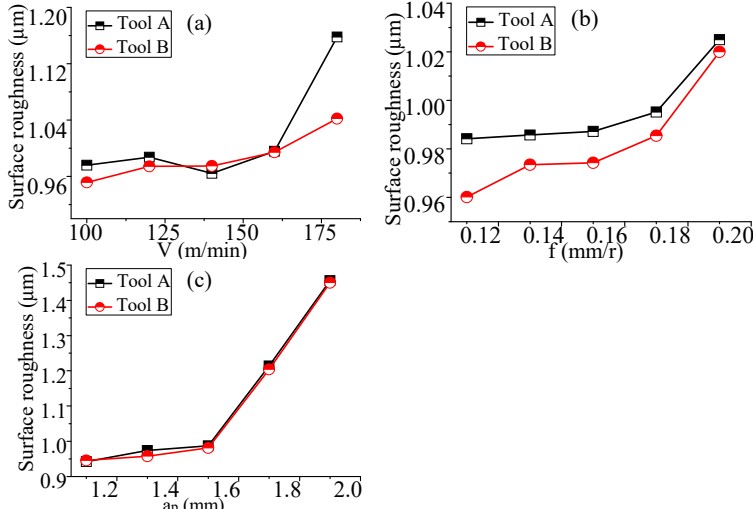

**Figure 8.** (**a**) Surface roughness changes with cutting speed, (**b**) surface roughness change with feed speed, (**c**) surface roughness changes with cutting depth range.

According to Figure 8b, the surface roughness of the workpieces cut by the tools A and B increases with the increase of the feed speed. The surface roughness of the workpiece cut by the tool B is lower. Figure 8c shows that the surface roughness of the workpieces cut by the tools A and B increase with the increase of the cutting depth rate, Under the same cutting conditions, the difference between the two surface roughnesses is very small.

In general, the cutting workpiece corresponding to tool B obtains lower surface roughness.

### 2.4. Analysis of Workpiece Hardenability

The phenomenon that the surface hardness of the workpiece increases during machining is called work hardening. Work hardening will increase the surface brittleness of the workpiece and reduce the impact resistance. At the same time, the hardening of the surface metal will make continuous processing difficult, and the tool wear is serious. In the cutting process of AISI 304, work hardening is a common phenomenon, which needs attention. The test process is shown in Figure 9.

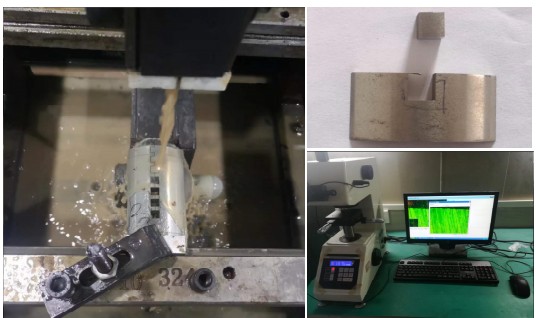

**Figure 9.** Workpiece hardness measurement.

The hardness of the AISI 304 bar matrix used in this experiment is 240 HV. The hardness change of the workpiece after the cutting experiment is shown in Figure 9. The degree of hardening is expressed by Formula (13).

$$N = \frac{H}{H_0} \times 100\% \tag{13}$$

where $H$ is the hardness value of the machined surface, and $H_0$ is the hardness of the matrix material.

As can be seen from Figure 10a, under the condition that the feed speed and cutting depth rate remain unchanged, the surface hardness of the workpiece by the tools A and B increase first, then decreases, and then increases rapidly with the increase of cutting speed. Under the same conditions, the surface hardness of the workpiece by the tool B is smaller than that of the tool A, and the degree of hardening is lower.

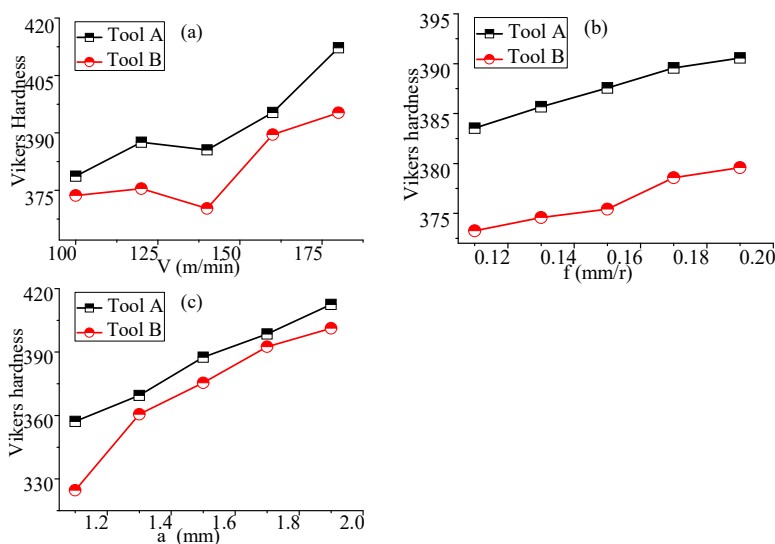

**Figure 10.** (**a**) Workpiece hardness changes with cutting speed, (**b**) workpiece hardness changes with feed speed, (**c**) workpiece hardness changes with cutting depth rate.

According to Figure 10b, the surface hardness of the workpiece by the tools A and B increase with the increase of the feed speed. Under the same conditions, the surface hardness of the workpiece by the tool B is smaller, and the degree of hardening is lower than the tool A. Figure 10c shows that the surface hardness of the workpiece by the tools A and B increase with the increase of the cutting depth rate. The surface hardness of the workpiece by the tool A is slightly higher than that of the tool B.

In general, the work hardening degree of the workpiece cut by tool B is lower.

## 3. Tool Durability Experiment and Analysis

### 3.1. Observation of Tool Surface Morphology

The above research shows that the tool B shows better cutting performance in different cutting parameter ranges. To demonstrate the performance of the tool in the actual whole cutting life stage, the paper designs an experiment on the cutting durability of the tools A and B, mainly observing and analyzing the surface morphology and element distribution of the tools. Based on the cutting test of tool durability and taking the wear of the flank face of the tool reaching a wideth of 0.15 mm as the tool blunt standard, tools A and tool B cut for 70 and 110 min, respectively.

An Olympus microscope was used to observe the morphology of the rake and flank face of the tools after cutting, as shown in Figure 11. It can be seen from the figure that when the wear width of the flank reaches 0.15 mm, the rake face of tool A is seriously worn, and the cutting edge has collapsed. The cutting edge of tool B remains relatively complete.

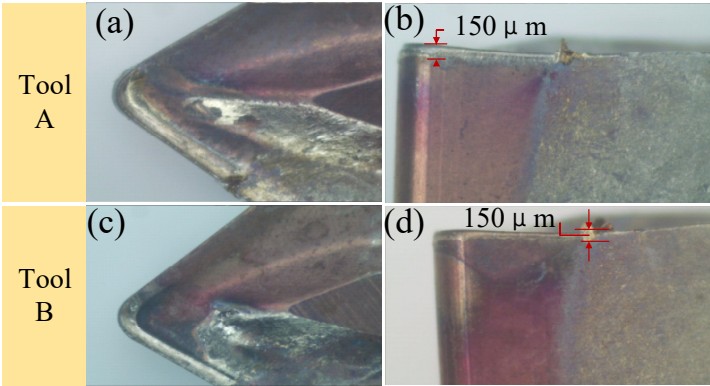

**Figure 11.** (**a**) The morphology of the rake face of tool A, (**b**) the morphology of the flank face of tool A, (**c**) the morphology of the rake face of tool B, (**d**) the morphology of the flank face of tool B.

AISI 304 has good toughness, high plasticity, low-thermal conductivity, and serious work hardening. In the cutting process, the tool–chip contact stress is very largem and the temperature is high; thus, abrasive wear, bonding wear, and oxidation wear easily occur on the rake and flank face of the tool.

After the cutting durability test, SEM and EDS analysis were carried out on the rake and flank face of tools A and B, as shown in Figure 11.

It can be seen from Figure 12 that the abrasive line on the rake face of tool A is obvious, and the vicinity of the cutting edge is white, indicating that the abrasive wear of the tool is serious, while the abrasive line of tool B is slight, the white area of the cutting edge is small, and the abrasive wear is small. From the flank face of tool A, the adhesive is very large, and the bonding wear is serious; however, the adhesion on the flank face of tool B is not obvious, and the bonding wear is relatively slight. From the microscopic morphology, it can be seen that the abrasive wear and bonding friction of tool B are softer than that of tool A.

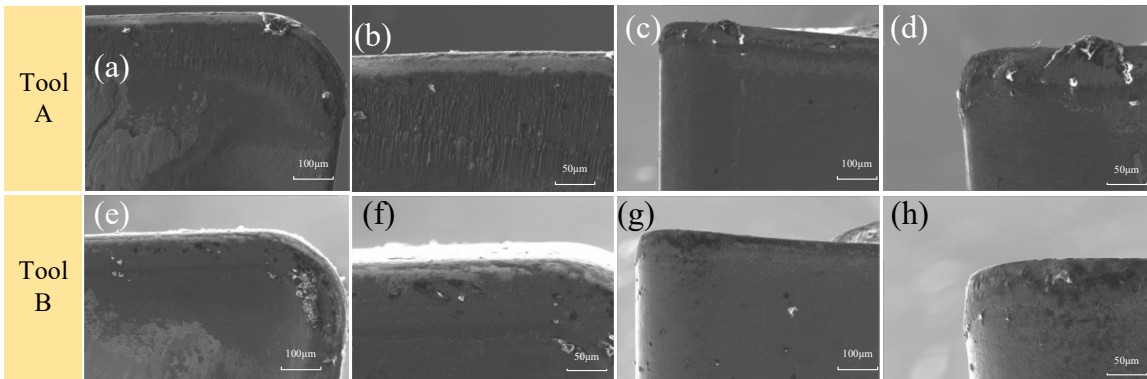

**Figure 12.** Micro characteristics of tools A and B. (**a**) Rake face of tool A, (**b**) Local enlarged view of rake face, (**c**) Flank face of tool B, (**d**) Local enlarged view of flank face, (**e**) Rake face of tool A, (**f**) Local enlarged view of rake face, (**g**) Flank face of tool B, (**h**) Local enlarged view of flank face.

*3.2. Element Energy Spectrum Analysis*

To accurately determine the status of tool wear in each area, the paper observed the element energy spectrum of the typical wear area of the rake face of the tool, as shown in Figure 13.

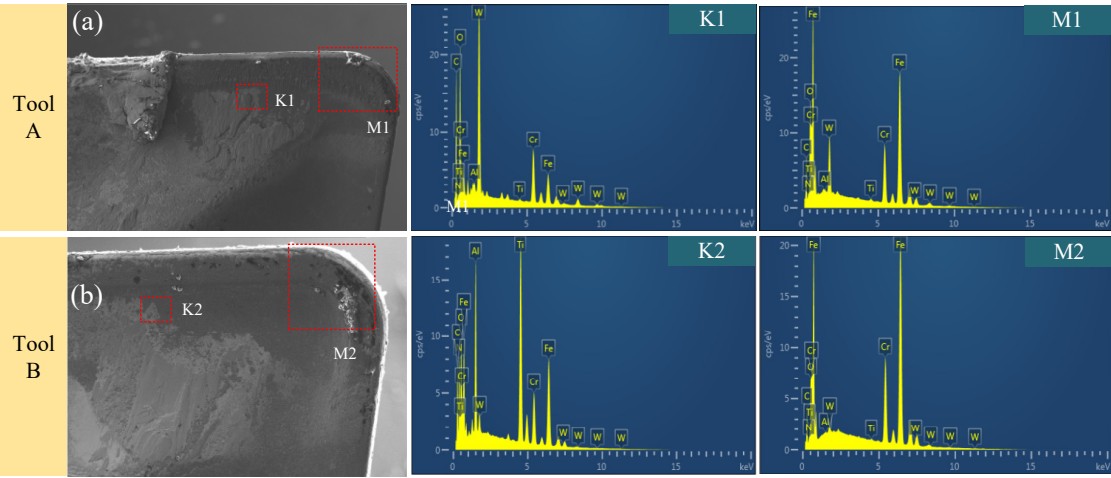

**Figure 13.** Energy spectrum analysis of elements of cutting tools A and B. (**a**) Rake face of tool A, (**b**) Rake face of tool B.

It can be seen from the figure that the content of the W element in the K1 area of tool A is the highest, followed by the O element. Element W is the tool base material, which indicates that the abrasive wear of tool A is serious, and the internal base material has been exposed, while element O comes from the air, where tool A has oxidative wear. The content of the K2, Al, and Ti elements in the similar area of tool B is higher, and the content of the O element is lower than that of tool A. The coating material of the tool is TiAlN, which indicates that the tool coating has not been polished, the abrasive wear is slight, and the oxidation wear is also slightly more.

It can be seen from Figure 14 that the line scan is carried out on the area where the cutting edge of the tools are seriously worn, and the element contents of Fe and O are observed. It is found from the comparison in the figure that the contents of Fe and O of tool B are low at the same position, which also shows that after the durability test, the bonding wear and oxidation wear of tool B are relatively slight.

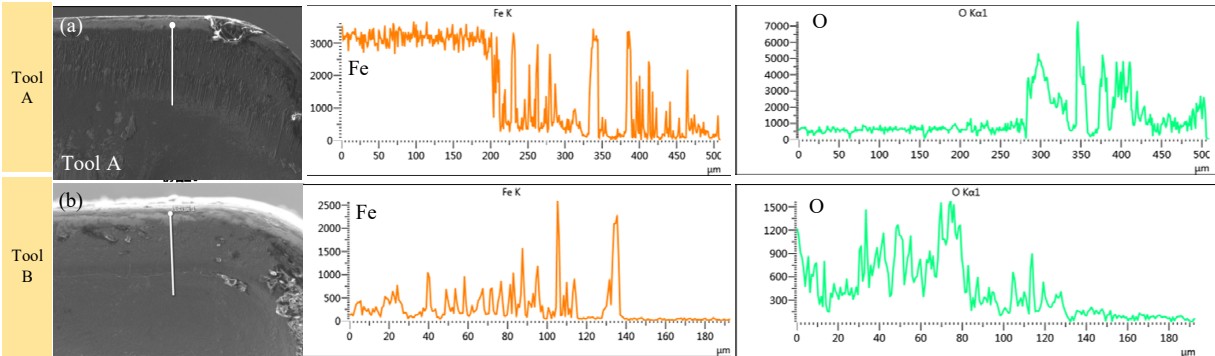

**Figure 14.** Fe and O concentration distribution of tool A and B line scanning. (**a**) Line scan of rake face of tool A, (**b**) Line scan of rake face of tool B.

## 4. Result and Discussion

Through the single factor cutting experiment, we find that the three-dimensional cutting force of tool B is smaller than that of tool A. The main cutting force and feed force decrease by more than 10%, and the shear energy decreases by more than 13%. The surface roughness of the workpiece cut by tool B is smaller, and the work hardening degree is lower. Through the durability test, it is found that the service life of the tool B is increased by 57%, and the adhesive wear and oxidation wear are less.

In the cutting process, the friction of the tool–chip contact interface is mainly composed of two parts: one is the internal friction zone near the tool tip, which is subjected to high temperature and high pressure, and the chip bottom plastic material is severely deformed and bonded with the tool front surface material, also known as the bonded friction zone. The second is that the tool–chip contact area is far away from the outer friction area of the tool tip. The pressure and temperature in this area are low, which obey the Coulomb's law of friction, which is called the sliding friction area.

It can be seen from Figure 15 that the placement of the micro groove of tool B changes the contact between the tool and the chip, and the actual contact length between the tool and chip is reduced. The length of the high-stress-bonding friction zone is about one-third of that of the tool A. At the same time, due to the width of the micro groove, the width of the sliding friction zone of tool B is only about one-half of that of tool A. Therefore, during the cutting process of tool B, the friction state between the tool and the chip is improved, the cutting heat and temperature is reduced, the cutting force is also reduced, and the bonding friction and oxidation wear of the tool are alleviated. At the same time, the existence of the micro groove makes the equivalent rake angle of the tool larger, the plastic deformation of the chip smaller, the thickness smaller, the cutting energy input smaller in the cutting process, and the cutting force and cutting temperature lower.

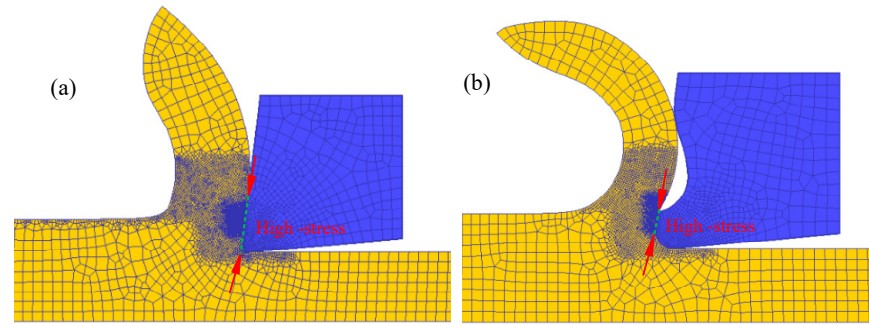

**Figure 15.** Simplified model of cutting tools A and B. (**a**) 2D model of cutting tool A, (**b**) 2D model of cutting tool B.

Due to the effect of the micro groove on the rake face, the effective contact time between the chip and the tool is reduced, and the heat generated by friction is reduced. In addition, the micro groove makes the chip obtain sufficient extension space, the chip deformation is reduced, and the chip deformation energy input is reduced.

## 5. Conclusions

During the process of cutting AISI 304, the tool is easy to wear, and the service life is short. To improve the cutting performance of the tool, a micro groove is designed on the rake face of the tool in this paper. The cutting single factor experiment and durability experiment were carried out, the cutting force, cutting energy, workpiece surface quality, tool wear element energy spectrum analysis in the cutting process were compared, and the experimental phenomena were analyzed. The specific conclusions are as follows:

1. Through the single factor cutting experiment, the three-dimensional cutting force of tool B is smaller than that of tool A. Among them, the cutting depth resistance decreases by more than 10%, the main cutting force decreases by more than 10%, the feed resistance decreases by more than 3%, and the shear energy decreases by more than 10%.
2. The design of the micro groove is based on the principle of reducing the contact zone between the tool and the chip and reducing the high-temperature area. Due to the effect of the micro groove of the tool B, the toolchip contact length is reduced, the cutting force is reduced, which leads to the reduction of the shear energy and the sliding energy, and the cutting temperature is decreased.
3. During the cutting process of tool B, the cutting force is smaller, and the cutting temperature is lower, which reduces the roughness and hardening degree of the workpiece surface after cutting.
4. Through the observation of surface morphology, the abrasive wear of the rake and flank face of tool B is lighter than that of tool A, and the wear marks are shallower. At the same time, the distribution of the main elements of the tool and the energy spectrum analysis of the local area show that the bonding wear and oxidation wear of tool B are also less than that of tool A.

High-speed turning of difficult to cut materials is a great challenge to cutting tools. In the next plan, the author will study the performance of high-speed turning stainless steel with micro groove tool.

**Author Contributions:** Conceptualization, Z.L.; methodology, Z.L and J.W.; writing—original draft preparation, Z.L.; validation, J.W. All authors have read and agreed to the published version of the manuscript.

**Funding:** This work was financially supported by the National Natural Science Foundation of China (Grant No.: 52105248), the science and technology top talent project of Guizhou Provincial Department of Education (Grant No.: [2022]086), the special funding of Guiyang science and technology bureau and Guiyang University (GYU-KY-[2022]).

**Institutional Review Board Statement:** Not applicable.

**Informed Consent Statement:** Not applicable.

**Data Availability Statement:** The authors confirm that the data supporting the findings of this study are available within the article.

**Conflicts of Interest:** The authors declare no conflict of interest.

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
