# Peer review of "Study on Cutting Performance of Micro Groove Tool in Turning AISI 304 and Surface Quality of the Workpiece"

_coatings, doi:10.3390/coatings12091326_

Round 1

Reviewer 1 Report

The authors applied the microgroove design of the tool for the machining of AISI 304 to analyze the mechanical properties and wear behaviors. In my opinion, this work is of interest. The paper contains enough material to be considered for a potential publication. I recommend its publication in this journal. However, the manuscript shortfall in the following aspects:

1- English should be improved in all the papers.

2- The abstract section should contain the objective, methods, results, and conclusions, with emphasis on the results and conclusions. The quantitative evaluation should be presented.

3- What is the novelty of the work?

4- The introduction section is written very well. However, it is too long that must be shortened.

5- Would you please indicate your critical reasons to choose the range of cutting parameters?  

6- The “Three-dimensional cutting force analysis” section is in poor condition. There is no information about boundary conditions, mesh element and mesh number for different parts, and so on.

7- The “Result” section must be created separately.

8- In general, the discussion is too much poor, regarding the huge literature about this issue.

9- A concise and factual conclusion is required. It must be summarized all achieved results. In the conclusion section, what is the science behind your work? What are the recommendations based on it?

Author Response

Letter of response

Manuscript number: coatings-1881441
Title: Study on cutting performance of micro groove tool in turning AISI 304 and surface quality of the workpiece.

Dear Editor(s) of coatings,
We are very glad to receive the reviewer’s comments on our manuscript. The reviewer’s comments are very helpful to improve our paper. We have revised the manuscript according to the reviewers’ comments, and the revised parts have been highlighted in red in the revised manuscript. Our responses to the reviewers’ comments are also shown below:

Response to Review 1:

Comment 1: English should be improved in all the papers.

Response 1: Thanks for your question. The paper will be polished as required.

Comment 2: The abstract section should contain the objective, methods, results, and

conclusions, with emphasis on the results and conclusions. The quantitative evaluation should be presented.

Response 2: Thanks for your question. We have revised the abstract of the paper. The details are as follows:(Line 10, 13-14, 17-18)

“In order to improve tool life, micro groove are designed on the rake face of the tool for the machining of AISI 304”

The shear energy is reduced by more than 13%the shear energy and friction energy are decreased, the surface roughness and the hardening degree of the workpieces are reduced.”

Through cutting experiments and theoretical analysis, the cutting performance of the micro groove tool has been improved.

Comment 3: What is the novelty of the work?

Response 3: Thanks for your question. The novelty of the work is based on three points: 1. The tool design based on the temperature field has a relatively complete theoretical basis. 2. It is the practical need of many enterprises to analyze the cutting performance of micro groove cutting tools with different cutting parameters. 3. In addition to the study of cutting force, cutting energy, tool wear and other general tool performance, the integrity of the surface quality of the workpiece in the cutting process is also analyzed.

Comment 4: The introduction section is written very well. However, it is too long that must be shortened.

Response 4: Thanks for your advice. We will revise it as required. We have deleted lines 41 to 47 and lines 53 to 55. This makes the introduction more concise.

Comment 5: Would you please indicate your critical reasons to choose the range of cutting parameters?

Response 5: Thanks for your question. This cutting parameter range is determined by the tool company through cutting experiments. Because the workpiece is difficult to cut, the interval between the values of cutting parameters is very small, which is conducive to accurately analyzing the performance of the tool. Some researchers use this method to study the cutting performance of the tool. (line 120-121)

“the paper designs the experiment of single factor cutting parameter changes of cutting AISI 304 with tools A and B.[32]

Jiang, H., He, L., Yang, X., Zou, Z.,Zhan, G. Prediction and experimental research on cutting energy of a new cemented carbide coating micro groove turning tool. The International Journal of Advanced Manufacturing Technology[J]. 89, 2335-2343 (2017).

Comment 6: The “Three-dimensional cutting force analysis” section is in poor condition. There is no information about boundary conditions, mesh element and mesh number for different parts, and so on.

Response 6: Thanks for your question. In the paper, the cutting force in the three directions of the tool is measured by the force measuring instrument Kistler 9257b. Since there is no cutting simulation, there is no need to set the boundary conditions, element grids, etc.

Comment 7: The “Result” section must be created separately.

Response 7: Thanks for your question. We added result in part 4 of the article. (line 352-357, line 378-381)

“Through single factor cutting experiment, we find that the three-dimensional cutting force of tool B is smaller than that of tool A. the main cutting force and feed force decrease by more than 10%, and the shear energy decreases by more than 13%. The surface roughness of the workpiece cut by tool B is smaller and the work hardening degree is lower. Through the durability test, it is found that the service life of the tool B is increased by 57%, and the adhesive wear and oxidation wear are less..”

“Due to the effect of micro groove on the rake face, the effective contact time between the chip and the tool is reduced, and the heat generated by friction is reduced. In addition, the micro groove makes the chip obtain sufficient extension space, the chip deformation is reduced, and the chip deformation energy input is reduced.”

Comment 8: In general, the discussion is too much poor, regarding the huge literature about this issue.

Response 8: Thanks for your advice and we added the content of "discussion". As follows: (line 378-381).

“Due to the effect of micro groove on the rake face, the effective contact time between the chip and the tool is reduced, and the heat generated by friction is reduced. In addition, the micro groove makes the chip obtain sufficient extension space, the chip deformation is reduced, and the chip deformation energy input is reduced.”

Comment 9: A concise and factual conclusion is required. It must be summarized all achieved results. In the conclusion section, what is the science behind your work? What are the recommendations based on it?

Response 9: Thanks for your advice and we made some modifications in the conclusion.(Line 390-398)

  1. Through single factor cutting experiment, the three-dimensional cutting force of tool B is smaller than that of tool A. Among them, the cutting depth resistance decreases by more than 10%, the main cutting force decreases by more than 10%, the feed resistance decreases by more than 3%, and the shear energy decreases by more than 10%.
  2. The design of micro groove is based on the principle of reducing between tool and chip contact zone and reducing high temperature area, due to the affect of the micro groove of the tool B, the tool chip contact length is reduced, the cutting force is reduced, which leads to the reduction of the shear energy and the sliding energy, and the cutting temperature is decreased.

Reviewer 2 Report

Dear authors, the manuscript ‘Scheme 304. and surface quality of the workpiece’, Manuscript ID: coatings-1881441, have some weaknesses that must be both responses and improved appropriately.

Please refer to the comments below:

1.      Some sentences, e.g. ‘In the practice of cutting production, within the range of cutting parameters recommended by tool manufacturers, each enterprise will choose different combinations of cutting parameters for cutting operations according to their own needs, which puts forward requirements for the cutting performance of tools in different combinations of cutting parameters’, lines 111-115, are too long and, correspondingly, difficult to follow for a (regular)reader. Please divide those sentences so that the reader cannot obtain what the author(s) are trying to convey.

2.      Were the formulas from 1.1 to 1.12 (and others in the manuscript), proposed by the author(s)? If not, the primary sources should be mentioned (cited). This can separate the novelty proposed from already published results.

3.      Considering surface roughness analysis in section 2.3., there is no word on details of the surface (topography) measurements. Moreover, both uncertainty and errors (noise) in the measurement process were not even mentioned. Please look for examples containing full responses and try to refer to both issues from surface metrology:

(1)   https://doi.org/10.1088/2051-672X/3/3/035004

(2)   https://doi.org/10.3390/ma14020333

(3)   https://doi.org/10.24425/mms.2021.137706

4.      In Figure 8, what is the ‘Surfaceness’? And, respectively, what unit is represented in this example?

5.      Moreover, which parameter is the ‘surface roughness’? In the sentence ‘According to figure 8(b), the surface roughness of the workpiece by the tool A and B increase with the increase of the feed speed, and the surface roughness of the workpiece by the tool B is smaller than the tool A under the same conditions. Figure 8(c) shows that the surface roughness of the workpiece by the tool A and B increase with the increase of the cutting depth rate, and the surface roughness of the workpiece by the tool A and B is relatively close.’, lines 253-258, it is difficult to follow.

6.      Most of the Figures should be enlarged so that is difficult to read. Sizes must go with the qualities as well.

7.      The ‘discussion’ section should contain the number 4, not 3. It is confusing. Moreover, all of the sections or, respectively, subsections should start with a capital letter.

8.      Some proposals for further studies should be presented in a separate section, e.g. ‘The Outlook’ or, respectively, in the ‘Conclusion’ section.

9.      What is the ‘reduced’ keyword? What does it indicate? It is confusing a lot.

10.  What the sentence ‘In recent years, many researchers have spent a lot of energy on research.’ presents? It looks too general and not required in the manuscript or, at least, improved to provide some information according to the review in the ‘Introduction’ section, information.

Moreover, the following editorial suggestion must be proposed as well:

11.  There should be space in line 34, ‘force[5]’.

12.  The dot in line 38, ‘Wang et al [9].’, is not required.

13.  But in line 42 the dot is required, ‘cutting, The’.

14.  Another space is missing in line 47, ‘Johannes et al.[13,14]’ and, respectively, in line 54, ‘[16]used’.

15.  There is no gap between Table 1 and further text.

16.  The sentence ‘at the same time’, in line 357, should start with a capital letter.

17.  Is the dot in line 369, ‘A and B. the cutting’, required?

18.  There is no gap between ‘Conclusions’, ‘Acknowledgements’ and ‘Conflict of Interest’.

19.  References should be unified according to the journal template requirements.

20.  The DOI links should be also added to the Reference list.

Generally, the proposed manuscript must be improved. Some issues make understanding the paper difficult and the reader confused. Therefore, the manuscript should be significantly improved before any further processing for the Coatings journal publication.

Author Response

Letter of response

Manuscript number: coatings-1881441
Title: Study on cutting performance of micro groove tool in turning AISI 304 and surface quality of the workpiece.

Dear Editor(s) of coatings,
We are very glad to receive the reviewer’s comments on our manuscript. The reviewer’s comments are very helpful to improve our paper. We have revised the manuscript according to the reviewers’ comments, and the revised parts have been highlighted in red in the revised manuscript. Our responses to the reviewers’ comments are also shown below:

Response to Review 2:

Comment 1: Some sentences, e.g. ‘In the practice of cutting production, within the range of cutting parameters recommended by tool manufacturers, each enterprise will choose different combinations of cutting parameters for cutting operations according to their own needs, which puts forward requirements for the cutting performance of tools in different combinations of cutting parameters’, lines 111-115, are too long and, correspondingly, difficult to follow for a (regular) reader. Please divide those sentences so that the reader cannot obtain what the author(s) are trying to convey.

Response 1: Thanks for your advice. We have modified the corresponding position and highlighted it in the original text. (line117-118).

“Within the range of cutting parameters, based on the needs of actual production, each enterprise will choose different cutting parameters.”

Comment 2: Were the formulas from 1.1 to 1.12 (and others in the manuscript), proposed by the author(s)? If not, the primary sources should be mentioned (cited). This can separate the novelty proposed from already published results.

Response 2: Thanks for your advice. First of all, this is not the original formula of the author. These formulas are classical theories and have authority in calculating cutting energy. Therefore, the paper add references. [Line 190-191; 204-206].
“According to metal cutting theory and relevant geometric knowledge[31]” In the process of cutting experiment, the shear energy and friction energy can be calculated by measuring the three-dimensional cutting force and chip thickness, combined with the known parameters and formulas 1-1 to 1-12[31].”
[31] Zou, Z. F., He, L., Jiang, H. W., Zhan, G.,Wu, J. X. Development and analysis of a low-wear micro-groove tool for turning Inconel 718. Wear[J]. 420, 163-175 (2019).

Comment 3: Considering surface roughness analysis in section 2.3., there is no word on details of the surface (topography) measurements. Moreover, both uncertainty and errors (noise) in the measurement process were not even mentioned. Please look for examples containing full responses and try to refer to both issues from surface metrology:
(1) https://doi.org/10.1088/2051-672X/3/3/035004
(2) https://doi.org/10.3390/ma14020333
(3) https://doi.org/10.24425/mms.2021.137706
Response 3: Thanks for your advice. We have added appropriate discussion in the paper. And references are cited. [Line 240-242].
“Surface roughness measurement and calculation need to pay attention to many factors, such as measurement noise, uncertainty evaluation. [32-34]
[32] Li, Z.,Gröger, S. Investigation of noise in surface topography measurement using structured illumination microscopy. Metrology and Measurement Systems[J]. 28 (2021).
[33] Podulka, P. Reduction of influence of the high-frequency noise on the results of surface topography measurements. Materials[J]. 14, 333 (2021).
[34] Haitjema, H. Uncertainty in measurement of surface topography. Surface Topography: Metrology and Properties[J]. 3, 035004 (2015).

Comment 4: In Figure 8, what is the ‘Surfaceness’? And, respectively, what unit is represented in this example?
Response 4: Thanks for your advice. We have changed ‘surfaceness’ to surface roughness. We have modified it in figure 8.(Line 246).

Comment 5: Moreover, which parameter is the ‘surface roughness’? In the sentence ‘According to figure 8(b), the surface roughness of the workpiece by the tool A and B increase with the increase of the feed speed, and the surface roughness of the workpiece by the tool B is smaller than the tool A under the same conditions. Figure 8(c) shows that the surface roughness of the workpiece by the tool A and B increase with the increase of the cutting depth rate, and the surface roughness of the workpiece by the tool A and B is relatively close.’, lines 253-258, it is difficult to follow.

Response 5: Thanks for your question. As shown in Fig. 8, the ordinate represents the surface roughness. We revised it again. [line 257-264].
“The surface roughness of the workpieces cut by the tools A and B increase with the increase of the feed speed. The surface roughness of the workpiece cut by the tool B is lower. Figure 8(c) shows that the surface roughness of the workpieces cut by the tool A and B increase with the increase of the cutting depth rate, Under the same cutting conditions, the difference between the two surface roughness is very small.”

Comment 6: Most of the Figures should be enlarged so that is difficult to read. Sizes must go with the qualities as well.
Response 6: Thanks for your question. The figures in the paper have been enlarged, as shown in Fig. 2 to Fig. 8.

Comment 7: The ‘discussion’ section should contain the number 4, not 3. It is confusing. Moreover, all of the sections or, respectively, subsections should start with a capital letter.
Response 7: Thanks for your question. The manuscript has been modified.

Comment 8: Some proposals for further studies should be presented in a separate section, e.g.‘The Outlook’ or, respectively, in the ‘Conclusion’ section.
Response 8: Thanks for your question. The manuscript has been modified.[Line 390-393].

“High speed turning of difficult to cut materials is a great challenge to cutting tools. In the next plan, the author will study the performance of high-speed turning stainless steel with micro groove tool.

Comment 9: What is the ‘reduced’ keyword? What does it indicate? It is confusing a lot.
Response 9: Thanks for your question. It has been changed in the corresponding position in the paper.[Line 19].
“Keywords: Micro groove tool; cutting force; energy; wear; surface quality”

Comment 10: What the sentence ‘In recent years, many researchers have spent a lot of energy on research.’ presents? It looks too general and not required in the manuscript or, at least, improved to provide some information according to the review in the ‘Introduction’ section, information.
Response 10: Thank you very much for your advice. This sentence has been deleted from the paper. [Line 29-30].

Comment 11: There should be space in line 34, ‘force[5]’.
Response 11: Thank you very much for your advice. It has been modified as required. [Line 37].

Comment 12: The dot in line 38, ‘Wang et al [9].’, is not required.
Response 12: Thank you very much for your question. It has been modified as required. [Line 41].

Comment 13: But in line 42 the dot is required, ‘cutting, The’.
Response 13: Thank you very much for your question. It has been modified in the paper.. [Line 47].

Comment 14: Another space is missing in line 47, ‘Johannes et al.[13,14]’ and, respectively, in line54, ‘[16]used’.
Response 14: Thank you very much for your question. It has been modified in the paper. [Line 50, 57].

Comment 15: There is no gap between Table 1 and further text.
Response 15: Thank you very much for your question. A row has been left blank after Table 1. [Line 102].

Comment 16: The sentence ‘at the same time’, in line 357, should start with a capital letter.
Response 16: Thank you very much for your question. It has been modified in the paper. [Line 370].

Comment 17: Is the dot in line 369, ‘A and B. the cutting’, required?
Response 17: Thank you very much for your question. It has been modified in the paper. [Line 386].

Comment 18: There is no gap between ‘Conclusions’, ‘Acknowledgements’ and ‘Conflict of Interest’.
Response 18: Thank you very much for your question. The row has been left at the corresponding positions. [Line 414, 419, 422].

Comment 19: References should be unified according to the journal template requirements.
Response 19: Thank you very much for your advice. References will be revised according to the journal format.

Comment 20: The DOI links should be also added to the Reference list.
Response 20: Thank you very much for your advice. The DOI links has been added to the Reference list.

Round 2

Reviewer 1 Report

The authors responded to the most of reviewer's comments for the original manuscript. Therefore, the publication is recommended.

Reviewer 2 Report

Dear authors, the manuscript ‘Scheme 304. and surface quality of the workpiece’, Manuscript ID: coatings-1881441, in the revised version, has been improved significantly so, respectively, can be further processed by the Editorial of the quality journal as the Coatings is.

Thank you for both your full responses and for taking into consideration all of the raised suggestions that, in their current form, were addressed properly.

From that matter, the manuscript was improved to be more suitable for publication, compared to the first version.

Concluding, from all your improvement, the manuscript, in its current, revised form, can be considered for publication in the Coatings journal.